# JOINT ROTATIONAL INVARIANCE AND ADVERSARIAL TRAINING OF A DUAL-STREAM TRANSFORMER YIELDS STATE OF THE ART BRAIN-SCORE FOR AREA V4

## ABSTRACT

Modern high-scoring models of vision in the brain score competition do not stem from Vision Transformers. However, in this paper, we provide evidence against the unexpected trend of Vision Transformers (ViT) being not perceptually aligned with human visual representations by showing how a dual-stream Transformer, a CrossViT *a la* Chen et al. (2021), under a joint rotationally-invariant and adversarial optimization procedure yields 2nd place in the aggregate Brain-Score 2022 competition (Schrimpf et al., 2020b) averaged across all visual categories, and at the time of the competition held 1st place for the highest explainable variance of area V4. In addition, our current Transformer-based model also achieves greater explainable variance for areas V4, IT and Behavior than a biologically-inspired CNN (ResNet50) that integrates a frontal V1-like computation module (Dapello et al., 2020). To assess the contribution of the optimization scheme with respect to the CrossViT architecture, we perform several additional experiments on differently optimized CrossViT's regarding adversarial robustness, common corruption benchmarks, mid-ventral stimuli interpretation and feature inversion. Against our initial expectations, our family of results provides tentative support for an *"All roads lead to Rome"* argument enforced via a joint optimization rule even for non biologically-motivated models of vision such as Vision Transformers.

## 1 INTRODUCTION

Research and design of modern deep learning and computer vision systems such as the NeoCognitron (Fukushima & Miyake, 1982), H-Max Model (Serre et al., 2005) and classical CNNs (LeCun et al., 2015) have often stemmed from breakthroughs in visual neuroscience dating from Kuffler (1953) and Hubel & Wiesel (1962). Today, research in neuroscience passes through a phase of symbiotic development where several models of artificial visual computation (mainly deep neural networks), may inform visual neuroscience (Richards et al., 2019) shedding light on puzzles of development (Lindsey et al., 2019), physiology (Dapello et al., 2020), representation (Jagadeesh & Gardner, 2022) and perception (Harrington & Deza, 2022).

Of particular recent interest is the development of Vision Transformers (Dosovitskiy et al., 2021). A model that originally generated several great breakthroughs in natural language processing (Vaswani et al., 2017), and that has now slowly begun to dominate the field of machine visual computation. However, in computer vision, we still do not understand why Vision Transformers perform so well when adapted to the visual domain (Bhojanapalli et al., 2021). Is this improvement in performance due to their self-attention mechanism; a relaxation of their weight-sharing constraint? Their greater number of parameters? Their optimization procedure? Or perhaps a combination of all these factors? Naturally, given the uncertainty of the models' *explainability*, their use has been carefully limited as a model of visual computation in biological (human) vision.

This is a double-edged sword: On one hand, perceptual psychologists still rely heavily on relatively low-scoring ImageNet-based accuracy models such as AlexNet, ResNet & VGG despite their *limited* degree of biological plausibility (though some operations are preserved, *eg.* local filtering, half-wave rectification, pooling). On the other hand, a new breed of models such as Vision Transformers has surged, but their somewhat non-biologically inspired computations have no straightforward mapping

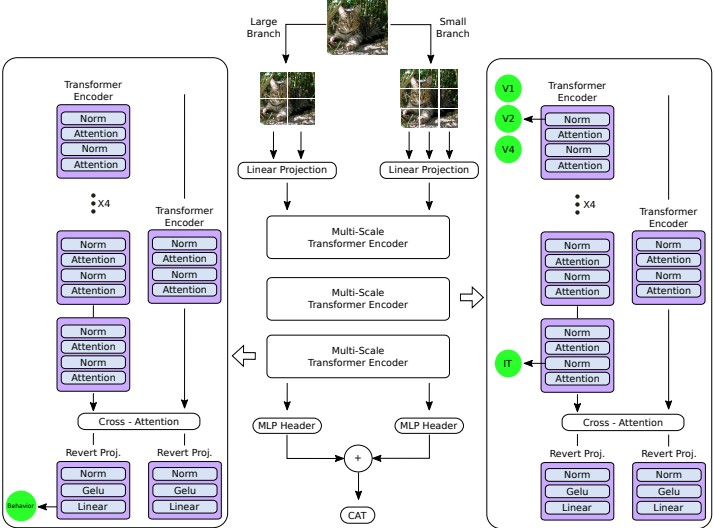

Figure 1: Diagram of CrossViT-18† (Chen et al., 2021) architecture & specification of selected layers for the V1, V2, V4, IT brain areas and the behavioral benchmark. Our Brain-Score 2022 competition entry was a variation of this model where the architecture is cloned, and the network is adversarially trained with hard data-augmentation rotations starting from a pre-trained ImageNet model.

to approximate the structure of the human ventral stream[1] – thus discarding them as serious models of the human visual system. Alas, even if computer vision scientists may want to remain on the sidelines of the usefulness of a biological/non-biological plausibility debate, the reality is that computer vision systems are still far from perfect. The existence of Adversarial examples, both artificial (Goodfellow et al., 2015; Szegedy et al., 2014) and natural (Hendrycks et al., 2021b), reflects that there is still a long way to go to close the human-machine perceptual alignment gap (Geirhos et al., 2021). Beyond the theoretical milestone of closing this gap, this will be beneficial for automated systems in radiology (Hosny et al., 2018), surveillance (Deza et al., 2019), driving (Huang & Chen, 2020), and art (Ramesh et al., 2022).

These two lines of thought bring us to an interesting question that was one of the motivations of this paper: *"Are Vision Transformers good models of the human ventral stream?"* Our approach to answer this question will rely on using the Brain-Score platform (Schrimpf et al., 2020a) and participating in their first yearly competition with a Transformer-based model. This platform quantifies the similarity via bounded [0,1] scores of responses between a computer model and a set of non-human primates. Here the ground truth is collected via neurophysiological recordings and/or behavioral outputs when primates are performing psychophysical tasks, and the scores are computed by some derivation of Representational Similarity Analysis (Kriegeskorte et al., 2008) when pitted against artificial neural network activations of modern computer vision models.

Altogether, if we find that a specific model yields high Brain-Scores, this may suggest that such flavor of Vision Transformers-based models obey a necessary but not sufficient condition of biological plausibility – or at least relatively so with respect to their Convolutional Neural Network (CNN) counter-parts. As it turns out, we will find out that the answer to the previously posed question is complex, and depends heavily on how the artificial model is optimized (trained). Thus the main contribution of this paper is to understand *why* this particular Transformer-based model when optimized under certain conditions performs vastly better in the Brain-Score competition achieving SOTA in such benchmark, and *not* to develop another competitive/SOTA model for ImageNet (which has shown to not be a good target Beyer et al. (2020)). The authors firmly believe that the former goal tackled in the paper is much under-explored compared to the latter, and is also of great importance to the intersection of the visual neuroscience and machine learning communities.

---

[1] Even at their start, the patch embedding operation is not obviously mappable to retinal, LGN, or V1-like primate computation.

| Rank | Model ID # | Description | Brain-Score | | | | | | $\rho$-Hierarchy |
| | | | Avg | V1 | V2 | V4 | IT | Behavior | |
|---|---|---|---|---|---|---|---|---|---|
| 1 | 1033 | Bag of Tricks (Riedel, 2022) [New SOTA] | **0.515** | **0.568** | **0.360** | 0.481 | 0.514 | **0.652** | -0.2 |
| 2 | 991 | CrossViT-18† (Adv + Rot) [Ours] | 0.488 | 0.493 | 0.342 | **0.514** | **0.531** | 0.562 | **+0.8** |
| 3 | 1044 | Gated Recurrence (Azeglio et al., 2022) | 0.463 | 0.509 | 0.303 | 0.482 | 0.467 | 0.554 | -0.4 |
| 4 | 896 | N/A | 0.456 | 0.538 | 0.336 | 0.485 | 0.459 | 0.461 | -0.4 |
| 5 | 1031 | N/A | 0.453 | 0.539 | 0.332 | 0.475 | 0.510 | 0.410 | -0.2 |

Table 1: Ranking of all entries in the Brain-Score 2022 competition as of February 28th, 2022. Scores in **blue** indicate **world record** (highest of all models at the time of the competition), while scores in **bold** display the highest scores of **competing entries**. Column $\rho$-Hierarchy indicates the Spearman rank correlation between per-Area Brain-Score and Depth of Visual Area (V1 $\rightarrow$ IT).

## 2    OPTIMIZING A CROSSVIT FOR THE BRAIN-SCORE COMPETITION

Now, we discuss an interesting finding, where amidst the constant debate of the biological plausibility of Vision Transformers – which have been deemed less biologically plausible than convolutional neural networks (as discussed in: URL_1 URL_2, though also see Conwell et al. (2021)) –, we find that when these Transformers are optimized under certain conditions, they may achieve high explainable variance with regards to many areas in primate vision, and surprisingly the highest score to date at the time of the competition for explainable variance in area V4, that still remains a mystery in visual neuroscience (see Pasupathy et al. (2020) for a review). Our final model and highest scoring model was based on several insights:

**Adversarial-Training**: Work by Santurkar et al. (2019); Engstrom et al. (2019b); Dapello et al. (2020), has shown that convolutional neural networks trained adversarially[2] yield human perceptually-aligned distortions when attacked. This is an interesting finding, that perhaps extends to vision transformers, but has never been qualitatively tested before though recent works – including this one (See Figure 3) – have started to investigate in this direction (Tuli et al., 2021; Caro et al., 2020). Thus we projected that once we picked a specific vision transformer architecture, we would train it adversarially.

**Multi-Resolution**: Pyramid approaches (Burt & Adelson, 1987; Simoncelli & Freeman, 1995; Heeger & Bergen, 1995) have been shown to correlate highly with good models of Brain-Scores (Marques et al., 2021). We devised that our Transformer had to incorporate this type of processing either implicitly or explicitly in its architecture.

**Rotation Invariance**: Object identification is generally rotationally invariant (depending on the category; *e.g.* not the case for faces (Kanwisher et al., 1998)). So we implicitly trained our model to take in different rotated object samples via hard rotation-based data augmentation. This procedure is different from pioneering work of Ecker et al. (2019) which explicitly added rotation equivariance to a convolutional neural network.

**Localized texture-based computation**: Despite the emergence of a *global* texture-bias in object recognition when training Deep Neural Networks (Geirhos et al., 2019) – object recognition is a compositional process (Brendel & Bethge, 2019; Deza et al., 2020). Recently, works in neuroscience have also suggested that *local* texture computation is perhaps pivotal for object recognition to either create an ideal basis set from which to represent objects (Long et al., 2018; Jagadeesh & Gardner, 2022) and/or encode robust representations (Harrington & Deza, 2022).

After searching for several models in the computer vision literature that resemble a Transformer model that ticks all the boxes above, we opted for a CrossViT-18† (that includes multi-resolution + local texture-based computation) that was trained with rotation-based augmentations and also adversarial training (See Appendix A.3 for exact training details, our *best* model also used $p = 0.25$ grayscale augmentation, though this contribution to model Brain-Score is minimal).

---

[2]Adversarial training is the process in which an image in the training distribution of a network is perturbed adversarially (*e.g.* via PGD); the perturbed image is re-labeled to its original non-perturbed class, and the network is optimized via Empirical Risk Minimization (Madry et al., 2018).

| Model ID # | Description | ImageNet (↑) Validation Accuracy (%) | Brain-Score (↑) Avg | V1 | V2 | V4 | IT | Behavior |
|---|---|---|---|---|---|---|---|---|
| N/A | Pixels (Baseline) | N/A | 0.053 | 0.158 | 0.003 | 0.048 | 0.035 | 0.020 |
| N/A | AlexNet (Baseline) | 63.3 | 0.424 | 0.508 | 0.353 | 0.443 | 0.447 | 0.370 |
| N/A | VOneResNet50-robust (SOTA) | 71.7 | **0.492** | **0.531** | **0.391** | 0.471 | 0.522 | 0.545 |
| 991 | CrossViT-18† (Adv + Rot) | 73.53 | 0.488 | 0.493 | 0.342 | **0.514** | **0.531** | **0.562** |
| 1084 | CrossViT-18† (Adv) | 64.60 | 0.462 | 0.497 | 0.343 | 0.508 | 0.519 | 0.441 |
| 1095 | CrossViT-18† (Rot) | 79.22 | 0.458 | 0.458 | 0.288 | 0.495 | 0.503 | 0.547 |
| 1057 | CrossViT-18† | **83.05** | 0.442 | 0.473 | 0.274 | 0.478 | 0.484 | 0.500 |

Table 2: A list of different models submitted to the Brain-Score 2022 competition. Scores in **bold** indicate the highest performing model per column. Scores in **blue** indicate **world record** (highest of all models at the time of the competition). All CrossViT-18† entries in the table are ours.

**Results:** Our best performing model #991 achieved 2nd place in the overall Brain-Score 2022 competition (Schrimpf et al., 2020b)) as shown in Table 1. At the time of submission, it holds the first place for the highest explainable variance of area V4 and the second highest score in the IT area. Our model also currently ranks 6th across all Brain-Score submitted models as shown on the main brain-score website (including those outside the competition and since the start of the platform's conception, totaling 219). Selected layers used from the CrossViT-18† are shown in Figure 1, and a general schematic of how Brain-Scores are calculated can be seen in Figure 2.

Table 3: Selected Layers of CrossViT-18†

| Benchmark | Layer |
|---|---|
| V1,V2,V4 | blocks.1.blocks.1.0.norm1 |
| IT | blocks.1.blocks.1.4.norm2 |
| Behavior | blocks.2.revert_projs.1.2 |

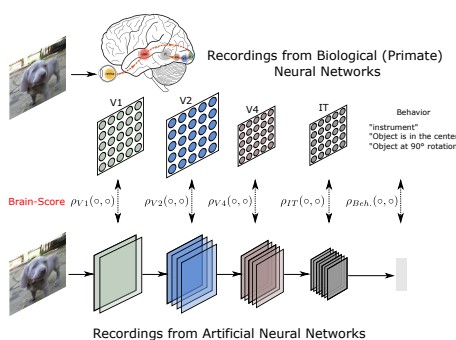

Figure 2: A schematic of how brain-score is calculated as similarity metrics obtained from neural responses and model activations.

Additionally, in comparison with the biologically-inspired model (VOneResNet50+ Adv. training), our model achieves greater scores in the IT, V4 and Behavioral benchmarks. Critically we notice that our best-performing model (#991) has a *positive* $\rho$-Hierarchy coefficient[3] compared to the new state of the art model (#1033) and other remaining entries, where this coefficient is negative. This was an unexpected result that we found as most biologically-driven models obtain higher Brain-Scores at the initial stages of the visual hierarchy (V1) (Dapello et al., 2020), and these scores decrease as a function of hierarchy with generally worse Brain-Scores in the final stages (*e.g.* IT).

We also investigated the differential effects of rotation invariance and adversarial training used on top of a pretrained CrossViT-18† as shown in Table 2. We observed that each step independently helps to improve the overall Brain-Score, quite ironically at the expense of ImageNet Validation accuracy (Zhang et al., 2019). Interestingly, when both methods are combined (Adversarial training and rotation invariance), the model outperforms the baseline behavioral score by a large margin (+0.062), the IT score by (+0.047), the V4 score by (+0.036), the V2 score by (+0.068), and the V1 score by (+0.020). Finally, even though is not the objective of our paper, our best model outperforms on Imagenet standard accuracy (73.53%) to a more biologically principled model such as the adversarially trained VOneResNet-50 (71.7%) (Dapello et al., 2020).

---

[3]$\rho$-Hierarchy coefficient: We define this as the Spearman rank correlation between the Brain-Scores of areas [V1,V2,V4,IT] with hierarchy: [1,2,3,4]

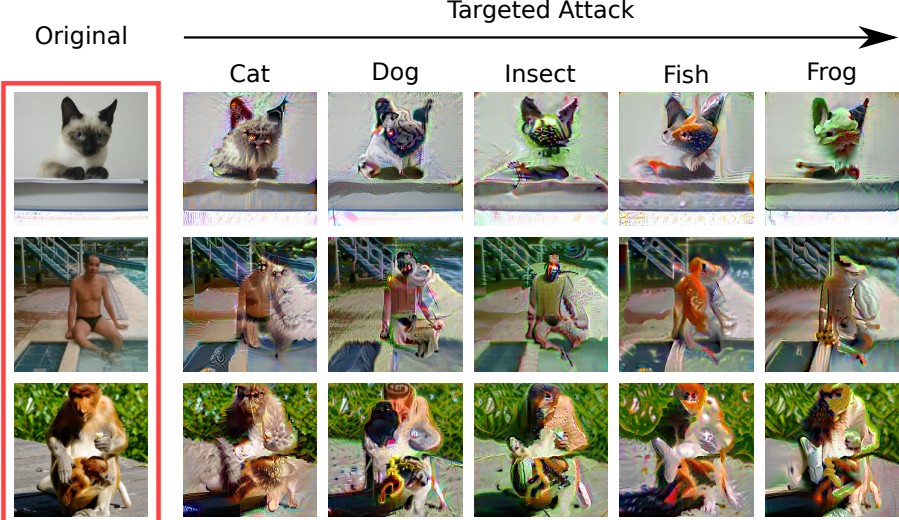

Figure 4: An extended demonstration of our winning model (CrossViT-18† [Adv. Training + Rot. invariance]) where a targeted attack is done for 3 images and the resulting stimuli is perceptually aligned with a human judgment of the fooled class. To our knowledge, this is the first time perceptually-aligned adversarially attacks have been shown to emerge in Transformer-based models.

## 3 ASSESSMENT OF CROSSVIT-18†-BASED MODELS

As we have seen, the *optimization* procedure heavily influences the brain-score of each CrossViT-18† model, and thus its alignment to human vision (at a coarse level accepting the premise of the Brain-Score competition). Since understanding the importance of each step in the constrained optimization procedure of the CrossViT is of vital importance not only when benchmarked with non human primate neurophysiological data but also in more classical in computer vision, We will now explore how different variations of such CrossViT's change as a function of their training procedure, and thus their learned representations via a suite of experiments. Additional experiments on common corruptions (ImageNet-C) and ImageNet-R can be seen in Appendix B.

### 3.1 ADVERSARIAL ATTACKS

One of our most interesting qualitative results is that the *direction* of the adversarial attack made on our highest performing model resembles a distortion class that seems to fool a human observer too (Figures 3, 4)[4]. Alas, while the adversarial attack can be conceived as a type of *eigendistortion* as in Berardino et al. (2017) we *find* that the Brain-Score optimized Transformer models are more perceptually aligned to human observers when judging distorted stimuli. Similar results were previously found by Santurkar et al. (2019) with ResNets, though there has not been any rigorous & unlimited time verification of this phenomena in humans similar to the work of Elsayed et al. (2018). Experimental details can be found in Appendix C

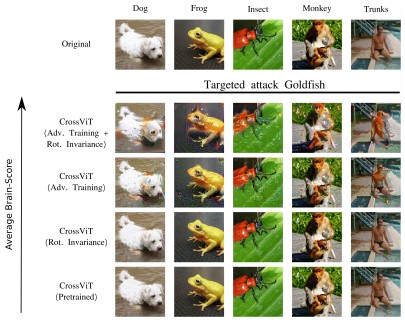

Figure 3: A qualitative demonstration of the human-machine perceptual alignment of the CrossViT-18† via the effects of adversarial perturbations. As the average Brain-Score increases in our system, the distortions seem to fool a human as well.

We also applied PGD attacks on our winning entry model (Adversarial Training + Rot. Invari-

---

[4]Rigorous psychophysical experiments are still needed to empirically show this, although see Santurkar et al. (2019); Feather et al. (2021); Harrington & Deza (2022) where attacks on brain-aligned models fool humans too.

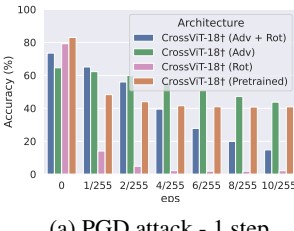 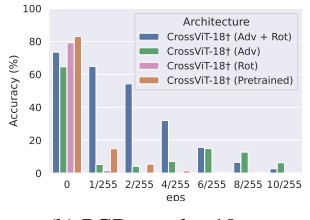 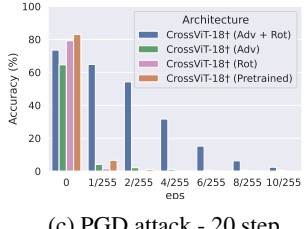

(a) PGD attack - 1 step      (b) PGD attack - 10 step      (c) PGD attack - 20 step

Figure 5: A suite of multiple steps [1,10,20] PGD-based adversarial attacks on clones of CrossViT-18†
models that were optimized differently. Here we see that our winning entry (Adversarial training
+ Rotation Invariance) shows greater robustness (adversarial accuracy) than all other models as the
number of steps of PGD-based attacks increases only for big step sizes of 10 & 20.

ance) on range $\epsilon \in \{1/255, 2/255, 4/255, 6/255, 8/255, 10/255\}$ and step-size = $\frac{2.5}{\#PGD_{iterations}}$ as
in the robustness Python library (Engstrom et al., 2019a) , in addition to three other controls: Adv.
Training, Rotational Invariance, and a pretrained CrossViT, to evaluate how their adversarial robust-
ness would change as a function of this particular distortion class. When doing this evaluation we
observe in Figure 5 that Adversarially trained models are more robust to PGD attacks (three-step size
flavors: 1 (FGSM), 10 & 20). One may be tempted to say that this is "expected" as the adversarially
trained networks would be more robust, but the type of adversarial attack on which they are trained is
different (FGSM as part of FAT (Wong et al., 2020) during training; and PGD at testing). Even if
FGSM can be interpreted as a 1 step PGD attack, it is not obvious that this type of generalization
would occur. In fact, it is of particular interest that the Adversarially trained CrossViT-18† with
"fast adversarial training" (FAT) shows greater robustness to PGD 1 step attacks when the epsilon
value used at testing time is very close to the values used at training (See Figure 5a). Naturally, for
PGD-based attacks where the step size is greater (10 and 20; Figs. 5b,5c), our winning entry model
achieves greater robustness against all other trained CrossViT's independent of the $\epsilon$ values.

## 3.2 MID-VENTRAL STIMULI INTERPRETATION

Next we are interested in exposing our models to a new type of stimuli/distortion class called
"texforms" because recent work has used these types of stimuli to show that human peripheral
computation may act as a biological proxy for an adversarially robust processing system (Harrington
& Deza, 2022), and that humans may in-fact use strong texture-like cues to perform object recognition
(in IT) without the specific need for a strong structural cue (Long et al., 2018; Jagadeesh & Gardner,
2022). Roughly speaking these texforms are very similar to their original counter-part, where they
match in global structure (*i.e.* form), but are locally distorted through a texture-matching operation
(*i.e. texture*) as seen in Figure 6 (Inset 0.).

In particular we are interested in knowing if the class separation is preserved across the visual
hierarchy for both the original and texform stimuli for the CrossViT-18† (Adv. Training + Rot.
Invariance but *not* for the CrossViT-18† (PreTrained) – as this would show a models' dissociation
modulated by a higher Brain-Score even if fixed with the same architecture. We used both the original
and texform stimuli (100 images per class) from Harrington & Deza (2022), showing only 2 classes
to the systems: primate and insect represented as circles and crosses respectively in Figure 6. In this
analysis, we will use a t-SNE projection with a fixed random seed across both models and stimuli to
evaluate the qualitative similarity/differences of their 2D clustering patterns.

We find that Pretrained CrossViT-18† models have trouble in early visual cortex read-out sections
to cluster both classes. In fact, several images are considered "visual outliers" for both original and
texform images. These differences are slowly resolved only for the original images as we go higher
in depth in the Transformer model until we get to the Behavior read-out layer. This is not the case for
the texforms, where the PreTrained CrossViT-18† can not tease apart the primate and insect classes
at such simulated behavioral stage. However, this story was to our surprise very different and more
coherent with human visual processing for the Adv + Rot CrossViT-18† where outliers no longer
exist – as there are none in the small dataset –, and the degree of linear separability for the original

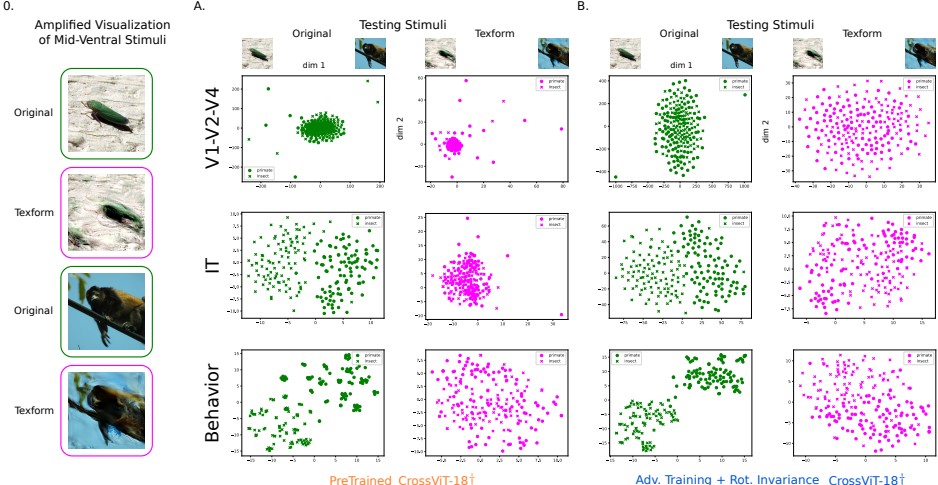

Figure 6: A comparison of how two CrossViT-18† models manage to classify original and texform stimuli. In (0.) we see a magnification of a texform, and in (A.,B.) we see how our winning Model Adv. + Rot. manages to create tighter vicinities across the visual stimuli, and ultimately – at the behavioral level – can separate both original and texform stimuli, while pretrained transformers seem to struggle with texform linear separability at the behavioral stage.

and texform stimuli increases in tandem through the hierarchy to near perfect separation for both stimuli at the behavioral stage.

### 3.3 FEATURE INVERSION

The last assessment we provided was inspired by feature inversion models that are a window to the representational soul of each model (Mahendran & Vedaldi, 2015). Oftentimes, models that are aligned with human visual perception in terms of their inductive biases and priors will show renderings that are very similar to the original image even when initialized from a noise image (Feather et al., 2019). We use the list of stimuli from Harrington & Deza (2022) to compare how several of these stimuli look like when they are rendered from the penultimate layer of a pretrained and our winning entry CrossViT-based model. A collection of synthesized images can be seen in Figure 7.

Even when these images are rendered starting from different noise images, Transformer-based models are remarkably good at recovering the structure of these images. This hints at coherence with the results of Tuli et al. (2021) who have argued that Transformer-based models have a stronger shape bias than most CNN's (Geirhos et al., 2019). We think this is due to their initial patch-embedding stage that preserves the visual organization of the image, though further investigation is necessary to validate this conjecture.

## 4 COMPARISON OF CROSSVIT VS VANILLA TRANSFORMER (VIT) MODELS

In this last section, we investigated what is the role of the architecture in our results. Did we arrive at a high-scoring Brain-Score model by virtue of the general Transformer architecture, or was there something particular about the CrossViT (dual stream Transformer), that in tandem with our training pipeline allowed for a more ventral-stream like representation? We repeated our analysis and training procedures with a collection of vanilla Vision Transformers (ViT) where we manipulated the patch size and number of layers with the conventions of Dosovitskiy et al. (2021) as shown in Figure 8.

Here we see that the Brain-Score on V2, V4, superior processing IT, Behavior and Average *increase* independent of the type of Vision Transformer used for our suite of models (CrossViT-18†, and multiple ViT flavors) except for the particular case of ViT-S/16 due to over-fitting (See Figure 11) that heavily reflects on the behavior score. To our surprise, adversarial training in some cases helped V1 score and in some not, potentially due to an interaction with both patch size and transformer depth that has not fully been understood. In addition, to our knowledge, this is also the first time that it has

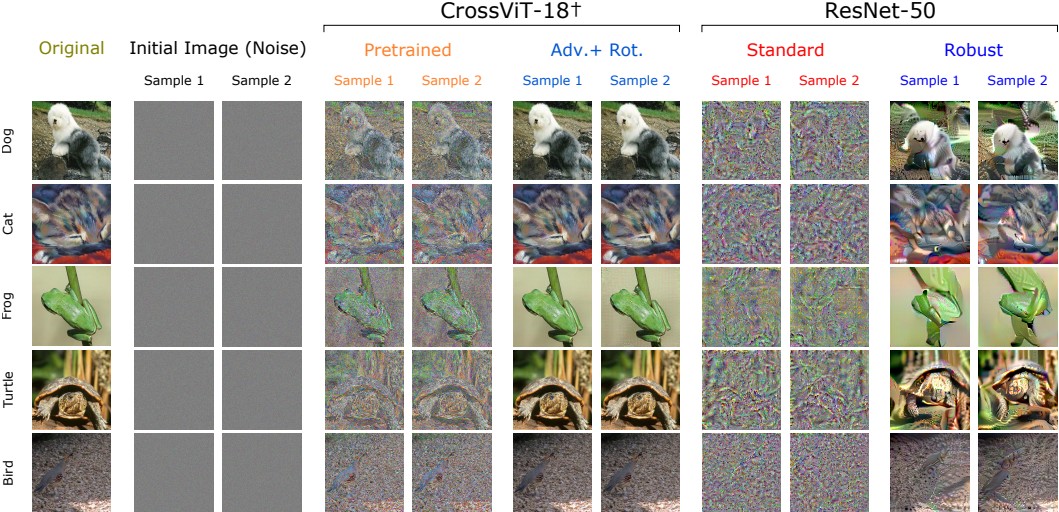

Figure 7: A summary of Feature Inversion models when applied on two different randomly samples noise images from a subset of the stimuli used in Harrington & Deza (2022). Standard and Pretrained models poorly invert the original stimuli leaving high spatial frequency artifacts. Adversarial training improves image inversion models, and this is even more evident for Transformer models. Notice that Transformer models independent of their optimization seem to preserve a higher shape bias as they recover the global structure of the original images. An extended figure can be viewed in the supplementary material.

been shown that adversarial training coupled with rotational invariance homogeneously increases brain-scores across Transformer-like architectures, as previous work has shown that classical CNNs (*i.e.* ResNets) increase Brain-Scores with adversarial training (Dapello et al., 2020).

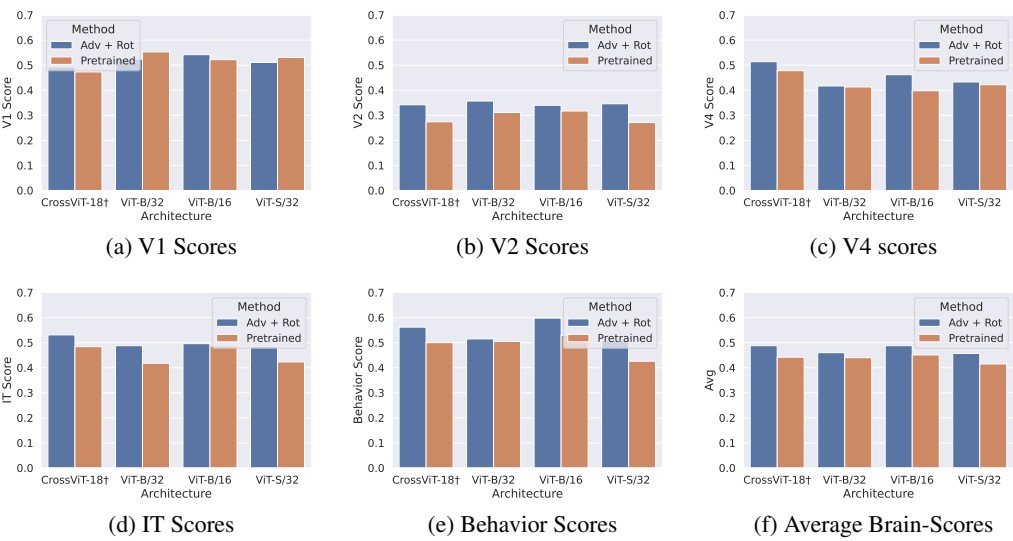

Figure 8: Similarity Brain-Score analysis on the different cortical areas of the ventral stream for vanilla transformers (ViT) and CrossViT. For nearly all Transformer variations, Adversarial Training with Joint Rotational Invariance increases per Area and Average Brain-Scores.

## 5 DISCUSSION

A question from this work that motivated the writing of this paper beyond the achievement of a high score in the Brain-Score competition is: How does a CrossViT-18† perform so well at explaining variance in primate area V4 without many iterations of hyper-parameter engineering? In this paper, we have only scratched the surface of this question, but some clues have emerged.

One possibility is that the cross-attention mechanism of the CrossViT-18† is a proxy for Gramian-like operations that encode local texture computation (vs global *a la* Geirhos et al. (2019)) which have been shown to be pivotal for object representation in humans (Long et al., 2018; Jagadeesh & Gardner, 2022; Harrington & Deza, 2022). This initial conjecture is corroborated by our image inversion effects (Section 3.3) where we find that CrossViT's preserves the structure stronger than Residual Networks (ResNets), while vanilla ViT's shows strong grid-like artifacts (See Figures 15, 16 in the supplementary material).

Equally relevant throughout this paper has been the critical finding of the role of the optimization procedure and the influence it has on achieving high Brain-Scores – even for non-biologically plausible architectures (Riedel, 2022). Indeed, the simple combination of adding rotation invariance as an implicit inductive bias through data-augmentation, and adding "worst-case scenario" (adversarial) images in the training regime seems to create a perceptually-aligned representation for neural networks (Santurkar et al., 2019).

On the other hand, the contributions to visual neuroscience from this paper are non-obvious. Traditionally, work in vision science has started from investigating phenomena in biological systems via psychophysical experiments and/or neural recordings of highly controlled stimuli in animals, to later verify their use or emergence when engineered in artificial perceptual systems. We are now in a situation where we have "by accident" stumbled upon a perceptual system that can successfully model (with half the full explained variance) visual processing in human area V4 – a region of which its functional goal still remains a mystery to neuroscientists (Vacher et al., 2020; Bashivan et al., 2019) –, giving us the chance to reverse engineer and dissect the contributions of the optimization procedure to a fixed architecture. We have done our best to pin-point a causal root to this phenomena, but we can only make an educated guess that a system with a cross-attention mechanism can *even under regular training* achieve high V4 Brain-Scores, and these are maximized when optimized with our joint adversarial training and rotation invariance procedure.

Ultimately, does this mean that Vision Transformers are good models of the Human Ventral Stream? We think that an answer to this question is a response to the nursery rhyme: *"It looks like a duck, and walks like a duck, but it's not a duck!"* One may be tempted to affirm that it is a duck if we are only to examine the family of in-distribution images from ImageNet at inference; but when out of distribution stimuli are shown to both machine and human perceptual systems we will have a chance to accurately assess their degree of perceptual similarity[5]. We can tentatively expand this argument further by studying adversarial images for both perceptual systems (See also Figure 9). Future images used in the Brain-Score competition that will better assess human-machine representational similarity should use these adversarial-like images to test if the family of mistakes that machines make are similar in nature than to the ones made by humans (See For example Golan et al. (2020)). If that is to be the case, then we are one step closer to building machines that can *see* like humans.

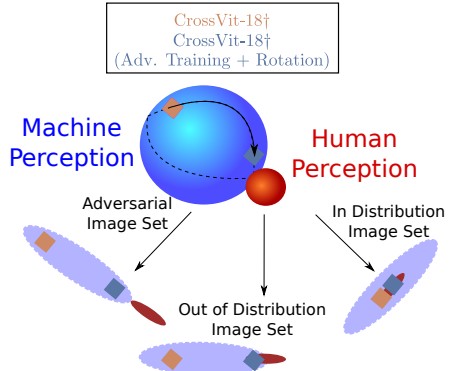

Figure 9: A cartoon inspired by Feather et al. (2019; 2021) depicting how our model changes its perceptual similarity depending on its optimization procedure. The arrows outside the spheres represent projections of such perceptual spaces that are observable by the images we show each system. While it may look like our winning model is "nearly human" it has still a long way to go, as the adversarial conditions have never been physiologically tested.

---

[5]Consider for example, that some stimuli used in Brain-Score are a basis set of Gabor filters, which are never encountered in nature

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

## A    EXPERIMENTAL SETUP

### A.1    DATASET

We used the ImageNet 1k (Deng et al., 2009) dataset for training. ImageNet1K contains 1,000 classes and the number of training and validation images are 1.28 million and 50,000, respectively. We validate the effectiveness of our models in the different datasets proposed in the Brain-Score (Schrimpf et al., 2020a) competition.

### A.2    CUSTOM SCHEDULER

The proposed learning rate scheduler is based on Jeddi et al. (2020) and is formulated as $LR = 0.00012 \times e - 0.0004$ for $e = 1$ and $LR = \frac{0.00002}{2^{e-2}}$ for $1 < e <= 6$. As shown in Figure 10, we start with a small learning rate and then it is smoothly increased for one epoch. We empirically found that fine-tuning the transformer for more than 1 epoch resulted in an under-fitting behavior of the adversarial robustness. After this first epoch, the learning rate is reduced very fast so that model performance converges to a steady state, without having too much time to overfit on the training data.

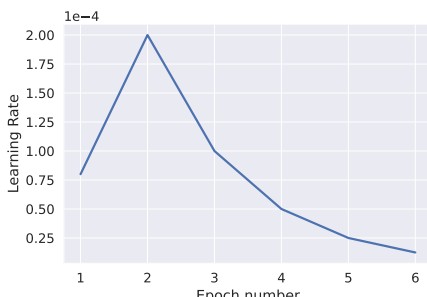

Figure 10: Custom scheduler used for training the Vision Transformer.

### A.3    TRAINING SETUP

We used a pretrained CrossViT-18† (Chen et al., 2021) downloaded from the timm library that is adversarially trained via a fast gradient sign method (FGSM) attack and random initialization (Wong et al., 2020). We opted for this strategy, known as "Fast Adversarial Training" as it allows a faster iteration in comparison with other common approaches (*e.g.* adversarial training with the PGD attack). In particular, all experiments used $\epsilon = 2/255$ and step size $\alpha = 1.25\epsilon$ as proposed originally in (Wong et al., 2020). However, in contrast to the previous method, we follow a 5 epoch fine-tuning approach with a custom learning rate scheduler in order to avoid underfitting. We optimize our networks with Adaptive Moment Estimation (Adam *a la* Kingma & Ba (2014)) and employed mixed precision for faster training. All input images were preprocessed with resizing to $256 \times 256$ followed

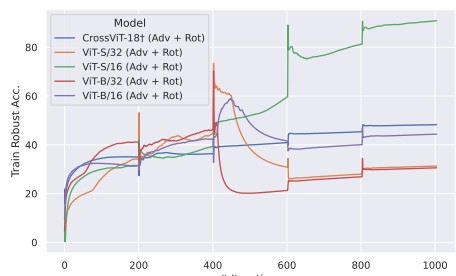

Figure 11: Training robust acc. of each Vision Transformer model (Adv + Rot). We clearly observed that ViT-S/16 has over-fitted during training.

by standard random cropping and horizontal mirroring. In the case of our best performing model (#991), we additionally incorporated a random grayscale transformation ($p = 0.25$) and a set of hard rotation transformations of (0°, 90°, 180°, 270°) – implicitly aiding for rotational invariance – due to the characteristics of images appearing in the behavioral benchmark of Rajalingham et al. (2018). All our experiments were run locally on a GPU-Tesla V-100. Each adversarial training of a vision transformer took around 48 hours.

# B    ADDITIONAL ASSESSMENT OF CROSSVIT-18†-BASED MODELS

## B.1    COMMON CORRUPTION BENCHMARKS

We also looked into how adversarial training would affect the performance of the different sets of neural networks to common corruptions that are *not* adversarial. To do this, we ran our models and benchmarked them to the ImageNet-C dataset (Hendrycks & Dietterich, 2019).

One would have expected Brain-Aligned models like our adversarially-trained + rotationally invariant CrossViT to also present strong robustness to common corruptions. To our surprise, this was not the case as seen in Table 4. This is a puzzling result, though there have been several bodies of work suggesting that adversarial robustness and common corruptions robustness are independent phenomena (Laugros et al., 2019), however Kireev et al. (2021) have proved otherwise contingent on the $l_\infty$ radius [6] – but now see Li et al. (2022).

| Network | Clean Accuracy (↑) | mce (↓) | Gauss | Shot | Impulse | Defocus | Glass | Motion | Zoom | Snow | Frost | Fog | Bright | Contrast | Elastic | Pixel | JPEG |
|---|---|---|---|---|---|---|---|---|---|---|---|---|---|---|---|---|---|
| ResNet50-Augmix | 77.53 | 67.1 | 65.5 | 65.1 | 66.4 | 67.7 | 81 | 63.9 | 65.5 | 71.6 | 70.9 | 66.5 | 57.8 | 60.2 | 76.9 | 59.5 | 68.5 |
| CrossViT-18† (Adv + Rot) | 73.53 | 79.5 | 80.7 | 81.6 | 83.2 | 90.2 | 78.7 | 82.4 | 80 | 77.6 | 74 | 107.9 | 65 | 100.4 | 74.2 | 57.4 | 58.7 |
| CrossViT-18† (Adv) | 64.60 | 88.8 | 85 | 85.7 | 86.7 | 96.7 | 88 | 92.1 | 91.3 | 85.8 | 83.6 | 109.3 | 82.2 | 104.9 | 90 | 70.3 | 80.9 |
| CrossViT-18† (Rot) | 79.22 | 73.1 | 75.4 | 76.7 | 75 | 75.7 | 85.3 | 72.3 | 79.2 | 68.8 | 70.9 | 64.3 | 54.7 | 67.6 | 78.4 | 75.4 | 76.4 |
| CrossViT-18† | **83.05** | **51** | **46.1** | **48.8** | **46.4** | **61.2** | **72.6** | **54.4** | **65** | **44.9** | **42.1** | **37.2** | **41.5** | **37** | **67.2** | **46.8** | **54.2** |

Table 4: A table showing the comparison of mean corruption errors (mce)'s across CrossViT models contingent on their training regime. A ResNet50-Augmix is shown as a reference of a particularly strong model to common corruptions. Here lower scores are indicative of better robustness to the different distortion types of Hendrycks & Dietterich (2019).

## B.2    IMAGENET-R

We also looked into how adversarial training would affect the performance of generalization to various abstract visual renditions. To do this, we ran our models and benchmarked them on the ImageNet-Rendition (ImageNet-R) dataset (Hendrycks et al., 2021a).

We observe that the accuracy on ImageNet-R decreases when the CrossViT is adversarially trained. However, when we combine the rotation invariance and adversarial training regimes, the accuracy on ImageNet-R becomes competitive with its pretrained version. In addition, we also appreciate that this combination does not affect the IID/OOD Gap with respect to the pretrained CrossViT.

| Network | ImageNet-200 (↑) | ImageNet-R (↑) | Gap (↓) |
|---|---|---|---|
| CrossViT-18† (Adv + Rot) | 90.75 | 41.14 | **49.61** |
| CrossViT-18† (Adv) | 85.52 | 35.73 | 49.79 |
| CrossViT-18† (Rot) | 93.89 | 37.35 | 56.54 |
| CrossViT-18† | 95.64 | **45.7** | 49.94 |

Table 5: A table showing the comparison of the accuracy on Imagenet-R dataset across CrossViT models contingent in their training regime.

## B.3    CENTER KERNEL ALIGNMENT TO UNDERSTAND CROSSVIT REPRESENTATIONS

We also calculated the center kernel alignment  (Kornblith et al., 2019) scores at each brain-region layer and on the Behavior and Inversion layers using a linear kernel. Besides, CKA scores were generated using the *'ImageNette'* validation dataset (Howard, 2019) which is a subset of 10 easily classified classes from ImageNet. The objective of this experiment is to understand how correlated are the variance of internal representations across the different versions of the optimized CrossViT-18†.

---

[6]Also see Li et al. (2022) that shows that generally robust models (robust to adversarial + commmon corruptions) have a preference for low-spatial frequency statistics.

We can see in Figure 12 that intermediate brain-region layers (IT, Behavior) tend to have similar representations across the 3 variants of CrossViT-18† (Rot. + Adv., Rot. and Adv.) based on the CKA score. In addition, we also appreciate that our best model (Crossvit-18† + Rot. + Adv.) is more correlated with their individual versions (Rot. and Adv.) than with its pretrained version.

It is also remarkable that at the penultimate layer of the largest branch (inversion layer), our best CrossViT possesses a very weak similarity with its pretrained form. This suggests that adversarial training and rotation invariance, either jointly or independently, strongly changes the representation of the final layers with respect to its pretrained version (CrossViT-18†).

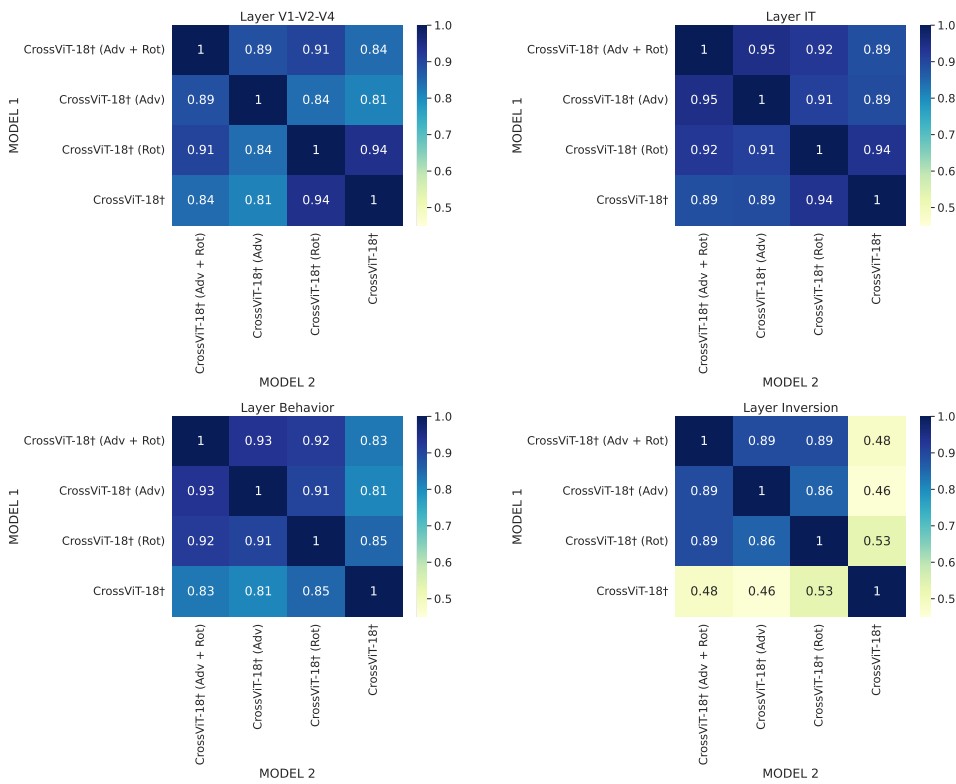

Figure 12: Similarity of representations at V1, V2, V4, Behavior and Inversion layers across the four versions of CrossViT-18† (pretrained, Adv., Adv. + Rot., Rot.). A score of 1.0 indicates highest representational similarity, while a score of 0.0 indicated lowest.

## C    ADVERSARIAL ATTACKS EXPERIMENTS

### C.1    TARGETED ADVERSARIAL ATTACKS

In this experiment, we maximize the probability of a specific class ("Goldfish" targeted attack) for the 4 flavors of the CrossViT-18†. We observed that as the average "Brain-Score" increases, the models tend to resemble more accurately the samples of the target class (Figure 3). In addition, we also performed targeted attacks for different classes on the ImageNet dataset as can be seen in Figure 4. Parameters used for these experiments can be found in Table 6

### C.1.1    ADVERSARIAL ROBUSTNESS TO PGD ATTACKS

Results of PGD adversarial attacks on different versions of CrossViT-18† can be found in Table 7. All experiments used $\epsilon \in \{1/255, 2/255, 4/255, 6/255, 8/255, 10/255\}$ and step-size $= \frac{2.5}{\#PGD_{iterations}}$ as in the robustness Python library (Engstrom et al., 2019a).

| Dataset | $\epsilon$ | Steps | Step size |
|---|---|---|---|
| ImageNet | 300 | 500 | 1 |

Table 6: Parameters used for the targeted attacks

| Model | $\epsilon - test(\uparrow)$ | | | | | |
|---|---|---|---|---|---|---|
| | 1/255 | 2/255 | 4/255 | 6/255 | 8/255 | 10/255 |
| CrossViT-18† (Adv + Rot) | 65.1/64.84/64.83 | 55.99/54.27/54.23 | 39.52/32/31.69 | 27.81/15.76/15.28 | 19.33/6.67/6.32 | 14.77/2.72/2.43 |
| CrossViT-18† (Adv) | 62.27/5.3/4.15 | 59.9/4.2/2.14 | 55.36/7.18/0.996 | 51.02/14.97/0.66 | 47.16/12.84/0.6 | 43.76/6.37/0.6 |
| CrossViT-18† (Rot) | 48/1.75/1.5 | 4.87/0/0 | 2.17/0/0 | 1.89/0/0 | 1.87/0/0 | 2.13/0/0 |
| CrossViT-18† | 48.31/14.87/6.64 | 44.01/5.56/1.1 | 41.58/1.47/0.09 | 40.96/0.59/0.02 | 40.79/0.35/0.01 | 40.9/0.13/0 |

Table 7: PGD adversarial attacks on different flavors of CrossViT-18†. Results represent adversarial accuracy at 1/10/20 PGD-iterations

# D BRAIN-SCORE

## D.1 METRICS

Brain-Score is a composite of multiple neural and behavioral benchmarks that score most of the artificial neural networks on how similar they are to the primate's brain mechanisms for core object recognitionSchrimpf et al. (2020a).

In the same direction, the Brain-Score competition was held for 4 months from December 21 to March 22. The objective was to evaluate models that engage with the whole ventral visual stream. These models were evaluated in 33 neuronal and behavioral benchmarks related to activity in macaque visual cortical areas V1, V2, V4, and IT and human psychophysical performance in a set of object classification tasks. The metrics used in the evaluation are the followings:

**Neural predictivity:** Measures how well the responses to given images in a model area predict the responses of a neuronal population of the corresponding area in the macaque brain. First, the model responses are mapped to the neuronal recordings using a linear transformation (PLS regression with 25 components) on a training set of images. Then the model's predictivity is determined for held-out images by computing the Pearson correlation coefficient between the model's predictions and the neuronal responses.

**Single-neuron property distribution similarity:** Measures whether single neurons in a model area are functionally similar to single-neurons in the corresponding monkey brain area. This is done by comparing the distribution of single-neuron response properties between the model area and the brain area using a similarity score (using the KS distance).

**Behavioral consistency:** Measures the behavioral similarity between the model and humans in core object recognition tasks. This metric does not measure the overall accuracy of the model but whether it can predict the patterns of successes and failures of humans in a set of object recognition tasks. Model's and humans' behavioral accuracies are first transformed to a d'statistic and then compared using the Pearson correlation coefficient.

### D.1.1 SELECTING BEST-BRAINSCORE LAYERS

Best performing layers on each vision transformer were selected by a brute-force approach. We evaluate each layer of the vision transformer models on each brain region and behavior dataset and select the layer that got the best score on the public benchmarks (in order to avoid overfitting) proportioned by Brain-Score organization. After this step, the "Adv + Rot" & pretrained versions of each transformer are submitted to the competition fixing best performing layers (See Table 8 ). We achieved our highest score at the time of our 4th submission, which was the lowest number of submissions in the competition (the winner of the competition performed nearly 60 submissions). All our results reflect the private scores obtained by each vision transformer model.

Additionally to the experiments on CrossViT-18†, we also evaluate the brain-scores on vanilla Vision transformers that can be seen in Table 9.

| Model | V1 | V2 | V4 | IT | Behavior |
|---|---|---|---|---|---|
| CrossViT-18† | blocks.1.blocks.1.0.norm1 | blocks.1.blocks.1.0.norm1 | blocks.1.blocks.1.0.norm1 | blocks.1.blocks.1.4.norm2 | blocks.2.revert_projs.1.2 |
| ViT-S/16 | blocks.1.mlp.act | blocks.3.attn.proj | blocks.3.norm2 | blocks.9.norm1 | pre_logits |
| ViT-S/16 | blocks.1.mlp.act | blocks.3.attn.proj | blocks.3.norm2 | blocks.9.norm1 | pre_logits |
| ViT-S/32 | blocks.1.mlp.act | blocks.10.norm1 | blocks.2.mlp.act | blocks.10.norm1 | pre_logits |
| ViT-B/16 | blocks.1.mlp.act | blocks.6.norm2 | blocks.2.mlp.act | blocks.8.norm1 | pre_logits |
| ViT-B/32 | blocks.1.mlp.act | blocks.6.norm2 | blocks.2.mlp.act | blocks.11.norm1 | pre_logits |

Table 8: Layers selected for each brain region on each vision transformer.

| Description | ImageNet(↑) | Brain-Score(↑) | | | | | |
|---|---|---|---|---|---|---|---|
| | Validation Acc. (%) | Avg | V1 | V2 | V4 | IT | Behavior |
| ViT-S/16 | 81.40 | 0.445 | 0.527 | 0.295 | 0.454 | 0.449 | 0.498 |
| ViT-S/32 | 75.99 | 0.415 | 0.531 | 0.271 | 0.422 | 0.423 | 0.426 |
| ViT-B/16 | 84.53 | 0.451 | 0.522 | 0.317 | 0.398 | 0.487 | 0.529 |
| ViT-B/32 | 80.72 | 0.440 | 0.553 | 0.311 | 0.413 | 0.418 | 0.505 |
| ViT-S/16 (Adv + Rot) | 50.44 | 0.443 | 0.506 | 0.332 | 0.470 | 0.496 | 0.409 |
| ViT-S/32 (Adv + Rot) | 55.20 | 0.457 | 0.512 | 0.347 | 0.433 | 0.485 | 0.508 |
| ViT-B/16 (Adv + Rot) | 67.25 | 0.486 | 0.536 | 0.332 | 0.470 | 0.496 | 0.598 |
| ViT-B/32 (Adv + Rot) | 53.01 | 0.457 | 0.524 | 0.357 | 0.417 | 0.472 | 0.515 |

Table 9: ImageNet accuracy, Brain-Scores of each brain area & Behavior benchmark evaluated on vanilla vision transformers. The spearman rank correlation between the validation accuracy and the average Brain-Score is $-0.28$ suggesting an *inverse* correlation between clean ImageNet accuracy and Brain-Score (Schrimpf et al., 2020a).

# E   IMAGE SYNTHESIS EXPERIMENTS

## E.1   STANDARD & ROBUST STIMULI

We used publicly available transformer models from timm library which were trained adversarially ($\epsilon = 2/255$ and step size $\alpha = 1.25$) as in (Wong et al., 2020) coupled with a set of hard rotation462 transformations of ($0°, 90°, 180°, 270°$) as proposed in this paper. In order to synthesize the standard and robust images, we used the penultimate layer (norm layer) in all of our vision transformer models except in the case of the CrossViT-18† versions in which we used the penultimate layer of the largest branch for all variations. Parameters used in these experiments can be seen in Table 10.

| Constraint | $\epsilon$ | Step-size | Iterations |
|---|---|---|---|
| $l_2$ | 1000 | 1 | 10000 |

Table 10: Parameters used for standard & robust stimuli by feature inversion

