# OpenReview forum: "Joint rotational invariance and adversarial training of a dual-stream Transformer yields state of the art Brain-Score for Area V4"
_ICLR.cc/2023/Conference — Submitted to ICLR 2023_

### Official Review · Reviewer_ywsh · 2022-10-20

**Confidence:** 5
**Correctness:** 4
**Technical Novelty And Significance:** 2
**Empirical Novelty And Significance:** 3
**Recommendation:** 5

**Clarity, Quality, Novelty And Reproducibility:**

Clarity: Paper is well structured and clearly written. Quality & Novelty: Authors' methodology puts together well-known adversarial training and data augmentation methods together to optimize a CrossViT, which yielded significantly high results in the Brain-Score competition, and the authors attempt to explain this observation. Hence I would say the technical novelty of this paper is limited in that sense. Reproducibility: Sufficient.


**Strength And Weaknesses:**

Strength: The manuscript is written clearly and in an understandable fashion. It achieves SOTA results in several categories at the time of their Brain-Score competition submission, and empirical ablations support their results.

Weaknesses: Currently the paper is restricted to explaining a set of experiments performed in the context of the competition, which appear to be outperformed to date. From a deep learning perspective, the methodological contribution is low since there is no clear architectural or algorithmic novelty proposed. This appears to be the main limitation in my opinion.

**Summary Of The Paper:**

The paper presents a dual-stream transformer architecture using CrossViT of [Chen et al. 2021], trained under a joint adversarial training objective with rotation invariance (imposed by hard rotation-based data augmentation), that achieved the second place in the BrainScore 2022 competition. Authors discuss in depth their optimization design choices empirically to illustrate how their ViT model could achieve high explainable variances in various competition metrics as an analogous model to the biological visual stream. Several ablation experiments with respect to the proposed joint optimization scheme are performed with the same CrossViT backbone, as well as comparisons with vanilla ViT models under the same training objectives.

**Summary Of The Review:**

I think the paper explores a very interesting problem in depth and demonstrates significant insights as to how simple adversarial training, data augmentation, and vision transformers could be combined to achieve high similarity to the biological visual stream in several metrics of the Brain-Score Competition. However I still believe the paper is not sufficiently strong at this venue without any independent novel methodological contribution. I have some specific questions to the authors summarized below:
- Explorations on adversarial training and robustness from Section 3.1: It would be reasonable to see the model thrive in more than the Brain-Score competition ranking, when it comes to claims on better adversarial robustness. For instance, to extend simpler PGD-type evaluations in Figure 5, authors could perform evaluations on AutoAttack and demonstrate how their adversarially trained CrossViT ranks in the Robustbench ImageNet category?
- In a similar vein, were there any explorations on the adversarial training pipeline (FAT, TRADES, standard AT etc.)? Can/did the authors explore alternative AT objectives for their model?
- How does the adversarially trained VOneNet [Dapello et al. 2020] compare in Figure 5?
- There have been a significant line of work on exploring adversarially trained vision transformers' generalization capabilities. Can the authors compare their baseline model with some of those?
- Authors' explorations in Supplementary Section B.1& B.2 are of major importance to the rest of the manuscript, even though the results look unsatisfactory. Their proposed optimization approach significantly hinders out-of-distribution generalization capability of CrossViT under common corruptions (even much lower than the ResNet-50 baseline). A discussion on this should be present in the main manuscript.
- Minor comment: Figure 8(a) legend is blocking the main results in the plot.

---

> ### Author Response · Authors · 2022-11-15
> **Comment response to Reviewer ywsh**
>
> Thank you for your careful review and your great questions Reviewer ywsh! We’re addressing several of the issues you raised below:
>
> > Weaknesses: Currently the paper is restricted to explaining a set of experiments performed in the context of the competition, which appear to be outperformed to date. From a deep learning perspective, the methodological contribution is low since there is no clear architectural or algorithmic novelty proposed. This appears to be the main limitation in my opinion.
>
> We’d like to clarify that the only Brain-Score related experiments that are proper from the benchmark are those shown in the first 2 pages (Table 1 and Table 2). All other experiments including all experiments in Section 3, and the out of distribution tests (ImageNet-R) and common corruptions robustness (see supplementary material), are numbers and baselines that are not part of the Brain-Score competition and were done on our side to disentangle the factors contributing to high brain score such as the optimization procedure and the architecture.
>
> And while our approach achieved 2nd place (of the overall average Brain-Score) in February 2022; our model has survived the test of time (albeit a short one), as we are currently placed 6th out of 217 submissions, and we currently rate 2nd in the V4 category out of these 217 submissions. **This is an objective quantitative assessment that we are in the world’s top 1% of computational models that explain non-human + human primate brain responses at the neurological and behavioral level.**
>
> You are indeed correct that the DL novelty in this paper is quite low, however, our scientific (rather than engineering-based) contribution relies on dissecting why these optimization procedures work so well for Brain-Score and comparing them across different transformer architectures as well. In the words of Reviewer kkk2 (score 8, confidence 5): **“Novelty: The topic under investigation is not novel, but the observations and conclusions that are brought forward - are. I think this work has an important place in the discussion about Brain-Score & brain-likeness in the context of artificial models of vision.”**
>
>
> > Re: additional baselines
>
> We think these are great experiments, but that in the space permitted in the paper will create a larger divergence from our main goal of understanding why a dual-stream transformer based model when optimized a particular way achieves high Brain-Scores. We opted for qualitative understanding rather than tables/numbers that provide very little insight.
>
> > Authors' explorations in Supplementary Section B.1& B.2 are of major importance to the rest of the manuscript, even though the results look unsatisfactory. Their proposed optimization approach significantly hinders out-of-distribution generalization capability of CrossViT under common corruptions (even much lower than the ResNet-50 baseline). A discussion on this should be present in the main manuscript.
>
> You are right, we’ve talked about this in the Discussion and will consider moving it to the main body. However, we think it’s worth stressing that having “bad” out of distribution generalization is not necessarily a weakness even if our model comes up top in the Brain-Score competition (it is a weakness in production, but it is not for scientific purposes, as this is a puzzling finding!!).
>
> For example, in Fel et al.’s recent Neural Harmonizer that was accepted at NeurIPS a couple of weeks ago, authors find that models that score high in ImageNet are not aligned with human perception (something that has been known in Visual Neuroscience for nearly a decade; Kriegeskorte & DiCarlo’s works from 2014 ; but largely ignored in CV/ML) -- such papers allow us to give a more principled understanding of DL systems applied to machine vision, rather than purely relying on tables and benchmarks even in the face of a new type of architecture. Our common corruptions results perhaps have a similar contribution to make: models that score high in brain-score need not to also have out of distribution robustness (and thus the brain-score image database should improve as we suggest in Figure 9!)
>
> Hopefully, our reply has clarified several of your lingering questions, and we’ve updated our manuscript correcting several of the typos/points you’ve raised. In light of the mixed reviews, let us know if there is something that is left to clarify that could increase your score. We think this paper if presented at ICLR will create many valuable discussions as it already has at this stage of reviewing and would thus be valuable for the community.
>
> > Minor comment: Figure 8(a) legend is blocking the main results in the plot.
>
> Thank you for your suggestion. We wanted to limit the "ylabel" so readers can appreciate the difference at each bar in the figure. Also the numerical values can be seen at the supplementary material section D.1.1 (Table 9)

---

> > ### Comment · Reviewer_ywsh · 2022-11-18
> > **Thanks to the authors for responses**
> >
> > Thanks to the authors for their time and responses to my review comments. To be fair, there is really not much additional input from the authors on my side that sparked different thoughts, unfortunately.
> >
> > I understand the authors' standpoint regarding the minimal technical novelty of their work. A response to the authors comment: "In the light of the mixed reviews, let us know if there is something that is left to clarify that could increase your score":
> >
> > Essentially that was what I discussed in my first round of revisions (score: 5, confidence: 5). Yet, the authors summarized the first four review bulletpoints in my original comments under "additional experiments", which are then discussed to be orthogonal to the main goal of the paper. In fact I did not think that way. The paper tries to explain the "high brain-score" one can obtain with a certain data augmentation & AT scheme with a specific vision transformer architecture. To that end I believe it would be more supporting to, for instance, explore today's architectural differences regarding this (point 4), or look into how do different AT algorithms that we nowadays believe to be state-of-the-art in "artificial network robustness" influence these results (point 2)? Even adding a "bio-plausible" model VOneNet to Figure 5 (point 3) should also not be a major concern.
> >
> > Overall I tend to keep my initial score the same.

---

> > > ### Author Response · Authors · 2022-11-18
> > > **Thank you + Reviewer Follow up**
> > >
> > > > Essentially that was what I discussed in my first round of revisions (score: 5, confidence: 5). Yet, the authors summarized the first four review bulletpoints in my original comments under "additional experiments", which are then discussed to be orthogonal to the main goal of the paper. In fact I did not think that way. The paper tries to explain the "high brain-score" one can obtain with a certain data augmentation & AT scheme with a specific vision transformer architecture. To that end I believe it would be more supporting to, for instance, explore today's architectural differences regarding this (point 4), or look into how do different AT algorithms that we nowadays believe to be state-of-the-art in "artificial network robustness" influence these results (point 2)? Even adding a "bio-plausible" model VOneNet to Figure 5 (point 3) should also not be a major concern.
> > >
> > > Hi Reviewer ywsh, these are very good points, and I think we both agree on the same things and experimental questions but we are interpreting the causes and effects in different ways. Our initial goal was to participate in the Brain-Score competition with a transformer-based model and make principled decisions to see if we could get a high Brain-Score. We did, and as a consequence discovered that things like adversarial training significantly boosted the score (though this is not new as proposed by Dapello, Marques et al [VOneNet] for CNNs) -- on our side the innovation lied on the Transformer model and exploring different transformer architectures too (Figure 8) and other augmentations (Tables 2 & 9) in a new neuroscience-based dataset (Brain-Score).
> > >
> > > But we agree: understanding what types of Adversarial training lends to improve brain-score is an important question, however circling back to our initial thoughts, this is a second-order question and not the main goal of this paper (and we think this deserves a second paper to be covered in its own right as it’s highly important!). In other words, our main goal was not to beat the Brain-score competition and win at all costs no matter the architecture or optimization regime (see the work of Reidel 2022 which is quite amazing too! And they did have such a goal, and did in fact win the competition), but rather as stated earlier, to explore and understand how we -- almost by accident -- obtained a high brain-score with such chosen transformer-based model and optimization scheme.
> > >
> > > It is ok if this rebuttal does not change your mind, or impression of our paper -- that is ok btw! Just thought we’d clarify where the initial goals came from as we wrote the paper, and that we really do value the insights you’ve provided to us that will help our future research directions.

---

### Official Review · Reviewer_kkk2 · 2022-10-24

**Confidence:** 5
**Correctness:** 4
**Technical Novelty And Significance:** 2
**Empirical Novelty And Significance:** 3
**Recommendation:** 6

**Clarity, Quality, Novelty And Reproducibility:**

Clarity: Apart from the detour (in my opinion) in Section 3, the paper is well structured, the ideas and the approach are clearly explained and allow to follow what is being demonstrated and how it was achieved.

Quality: High

Novelty: The topic under investigation is not novel, but the observations and conclusions that are brought forward - are. I think this work has an important place in the discussion about Brain-Score & brain-likeness in the context of artificial models of vision.

Reproducibility: The steps taken are clearly described and should be reproducible. Making the code publicly available would be of great help.

**Strength And Weaknesses:**

The narrative of this work has an unfortunate break when the Section 3 starts: the first four pages very clearly outline the premises and ideas of the authors and the scientific question emerges (being something like "can transformers score high and what would it take and what that would mean"). In Section 2 the main answer is given -- the fact that properly "motivated" transformer scores high on Brain-Score. But the the transition to Section 3 is unclear, it makes the reader feel that Section 3 is where the main results is, while actually Section 3 is a collection of "optional" observations. I would recommend to explain *why* you are now going to "explore how different variations of such CrossViT’s change as a function of their training procedure" and in each of those sections bring forward what does this analysis do for the main claim of the paper or how does it contribute to the "brain-likeness discussion".

Currently some of the points you make in Section 3 are dangling unconnected to the rest. For example model's ability or inability to classify between 2 classes in original vs texform space. Why is that relevant? Do we (humans) really use texture-like cues to perform object recognition without the need for shapes? I would disagree with that already on the bases of my personal experience -- looking at Figure 6, panel 0, texform insect -- without context I would have never classified that as an insect? I think we (humans) do need the shape quite a lot. So the fact that one model can perform well on textures while another cannot does not convince in that model's brain-likeness.

Section 3.3 feels even more disconnected from the whole brain discussion. It basically just states that transformers have stronger reliance on shape information. Ok, it's a cool fact to know, but how does that connect to the topic of the paper? And how does this finding relate to the one presented in 3.2 where the whole point was somewhat the opposite -- that high Brain-Scoring model can work well in texture-form?

Taken together sections 3.1 to 3.3 take the reader away from the main conversation and into some vague and as it seems irrelevant observations that blur the excitement and clarity of the message in Section 2.

Addressing this broken narrative I think will do a great service to bring forward the numerous strengths of this paper:
- clear and bold scientific question
- logical exploration and experimental work to find the answer
- important message for the bioplausible vision community
- clear language

**Summary Of The Paper:**

The authors challenge the common notion that transformer-like architecture is not particularly brain-like because, to the best of our neuroknowledge, the self-attention mechanism employed in transformers is not inspired by how the brain works. To that end they pick a few optimization constraints that in the past have shown to increase the Brain-Score (a metric that measures how close a certain ANN model's activity is close to brain activity is in representational space) and train a transformer architecture subject to all these constrains. And, as they discover, when trained like this, a transformer architecture is no worse that other ones, more "bioplausible" artificial models of vision and it scores even higher than some architectures that were manually crafted to be brain-like.


**Summary Of The Review:**

This is an interesting paper and a valuable contribution in that it should spark an important discussion on how should we go about assessing brain-likeness of an ANN architecture. To me the most important message from this paper is, as I've long suspected, that if you fiddle with any architecture long enough then you can make it score high on Brain-Score. We, as a community, should be much more careful when claiming brain-likeness based on Brain-Score, and this paper is an important voice in this discussion.

The paper is well-written, the experiments support the claims and those claims are important and/or novel.

My recommendation is to accept this work as I would definitely like to see it presented at the conference.


------ UPDATE after having a joint call with other reviewers ------

The way I understood the main contribution of this work initially was to highlight that with enough effort one can make even a transformer-based architecture to score on brain-likeness test. The main claim of the paper as I saw it then appeared to me something like "let's not put too much trust in such way of evaluating brain-likeness".

However, after reading the final version and looking at the text with the perspective other reviewers provided, I can see that this is not the main claim of the work and is a secondary message (or, it does not come out of the manuscript clearly enough). The main claim of the work becomes (as the title says) that with a certain set of add-ons a vision transformer scores high. And this, while still curious, to me does not constitute as important of a message as the one above might. In order to advance knowledge in NS this would need to have one step further, to establish a deeper connection between the evaluated add-ons and brain structure/function (and I think it was never the claim that this work advanced the knowledge in ML per se). Leaving us with a set of interesting observations, and the (surprising? or not?) outcome that it scores high on BrainScore.

My general recommendation is still to accept, but now with a score of 6 instead of 8 (there's no "7" available).

---

> ### Author Response · Authors · 2022-11-15
> **Comment response to Reviewer kkk2**
>
> Thank you for your enthusiasm in your review, Reviewer kkk2! We will be clarifying several of the points you made in your review below.
>
> > I would recommend to explain why you are now going to "explore how different variations of such CrossViT’s change as a function of their training procedure" and in each of those sections bring forward what does this analysis do for the main claim of the paper or how does it contribute to the "brain-likeness discussion".
>
> As mentioned in the response to Reviewer 84qC: “Our analysis suggests that the optimization procedure is really important for generating good models of the human ventral stream based on the Brain-Score Metric which quantifies the similarity of neural network responses between a computer vision model (artificial) and a set of non-human primates (biological). In that sense, we believe that understanding the importance of each step in the constrained optimization procedure of the CrossViT is of vital importance since stacking multiple data augmentations (Alexander Riedel 2022) will probably increase the “brain-likeness” but we would not understand which aspects really matters in this increase. Additionally, It is also important to assess the performance of these effects not only when benchmarked with non-human primate neurophysiological data (Brain-Score) but also in more classical computer vision like:
> * Adversarial Robustness under PGD attacks
> * Feature Inversion which has shown that aligned models with human visual perception in terms of their inductive biases and priors will show renderings that are very similar to the original image even when initialized from a noise image
> * Mid-Ventral stimuli interpretation for higher levels of visual processing in artificial NN models
> * Out of distribution performance and robustness when evaluated via common corruptions
>
> > Regarding cohesion on Sections 3.2 and 3.3
>
> **This is a great observation from Reviewer kkk2**. However, there may have been a lack of clarity on our behalf of what the “texforms” are, which aligns the results from both sub-sections. Texforms, are stimuli that preserve global shape/structure, and also preserve local texture statistics. In other words, texforms are not a globally texture-matched stimuli like in Gatys et al. or Portilla & Simoncelli (PS) -- but rather stimuli that are rendered with that same texture synthesis procedure at a local level, and thus implicitly & coarsely persevering the structure of the reference stimuli. To be more precise: imagine dividing an image into a 4x4 grid and performing local PS texture synthesis on each patch -- with the additional constraint of doing low-spatial frequency matching of the reference image; that is a texform.
>
> That being said, since texforms preserve local texture and global coarse shape, there is coherence with the results of Subsections 3.2 and 3.3 (near perfect image inversion). In addition, the evidence and support that humans do use global shape and local texture encoding mechanisms for object processing was recently discovered in both the works of Long, Yu & Konkle (PNAS 2019) and even more recently in Jagadeesh & Gardner (PNAS 2022). This result for Transformers has only been explored from another computational approach in Tuli et al. (CogSci 2021) -- we have added this reference in our updated manuscript (!)
>
> **Also see that we re-wrote section 3.2 to make things clearer as suggested by Reviewer 2i2E (also accept).** We hope this has provided more clarity to your inquiry, we’ve updated our manuscript to reflect this. Please let us know if it requires even more clarity as this is indeed quite an important point that should not evoke confusion in the readers.
>
> Finally, thank you for your thoughts and reviews on our paper Reviewer kkk2, we are just as enthusiastic as you with regards to tentatively sharing our results with the ICLR community!

---

### Official Review · Reviewer_2i2E · 2022-10-24

**Confidence:** 4
**Correctness:** 3
**Technical Novelty And Significance:** 1
**Empirical Novelty And Significance:** 3
**Recommendation:** 8

**Clarity, Quality, Novelty And Reproducibility:**

Clarity:

Overall the manuscript is clear.

In the abstract : I don't know what is this "all roads lead to Rome" argument... The authors should be more specific.

In section 3.1: Perceptual alignment between an image and the attacked class is unsupported by any data. It is only a qualitative claim from the authors.

In section 3.2: the relation with peripheral vision, the use of texform stimuli and the reasons why they are used are unclear. The first sentence of the section is misleading as your are not really interested in how well the classes are getting separated when going up in the hierarchy but whether class separation is preserved when using texform (the locally texture-matched version of the image). I think you should start this section with what is stated in the 2nd paragraph (texture-like recognition in human => mid-ventral visual cortex and texture / peripheral vision => robust perception => testing that using natural images and texform and comparing classificiation).

Fontsize in figures should be close to the main text fontsize.

For figure 6: I suggest to use two colors for the two classes (the marker are hard to distinguish and this what is important). You can add a large frame of the right color around each column to highlight the original vs texform stimuli.



Quality:

Good.



Novelty:

Vision Transformers are not yet widely tested against biological vision datasets.



Reproducibility:

Through brain-score competition

**Strength And Weaknesses:**

Strength:
- high performance of vision transformer for predicting neural activity (brain score competition),
- using an architecture that incorporate biologically plausible features (multiscale, invariance, texture features),
- an attempt to explain the performances (robustness, texture-like computation, invertibility),
- include models comparison

Weaknesses:
- the fact that targeted attacks result in stimuli that are perceptually aligned with human judgement is not supported by any perceptual experiment

**Summary Of The Paper:**

The authors test a neural network (NN) architecture based on vision transformers on the brain-score competition. When using adversarial training (gradient attack) and rotated data augmentation, the NN reaches state of the art performance for area V4. In contrast with other best performing models, their model is getting better and better at predicting higher visual area. Then, the authors assess the variation of training to highlight that robustness probably explains the performance ranking of the different training methods. They also draw a parallel peripheral vision to explain specifically the good performance in area V4 and show that their model is invertible. Finally, a last comparison is done against classical vision transformers.

**Summary Of The Review:**

Sadly, the brain-score competition is not part of a conference with proceedings... This would be where this paper belongs to.
I am inclined to accept this paper once the authors have accounted for my comments.

---

> ### Author Response · Authors · 2022-11-15
> **Comment response to Reviewer 2i2E**
>
> Thank you for your time and energy into reviewing our paper Reviewer 2i2E! We have made the modifications you suggested to our main manuscript. As per your additional concerns, we hope by answering these comments, you are encouraged to raise your score.
>
> > the fact that targeted attacks result in stimuli that are perceptually aligned with human judgment is not supported by any perceptual experiment
>
> > In section 3.1: Perceptual alignment between an image and the attacked class is unsupported by any data. It is only a qualitative claim from the authors.
>
> You are right, these are author opinions, and are not from an actual psychophysical study as those done in Harrington & Deza (ICLR, 2022) or in Feather et al. (NeurIPS 2019). However, previous works for example by Santurkar et al. (NeurIPS 2019), also suggest that these types of distortions fool humans -- despite no psychophysical evidence. However, we agree & psychophysical experiments should be done on these stimuli, and this is current follow-up work from this paper [see for example the analysis on time-limited display of adversarial images on humans by Elsayed et al., (NeurIPS 2019)]. **We have clarified this in our manuscript w/ a footnote.**
>
> > I don't know what this "all roads lead to Rome" argument... The authors should be more specific.
>
> Authors mean that there might be different methods (or combinations of them) in order to generate a new set of brain-aligned vision models (or what look like them) including modifications in the architecture, training recipe, training data, etc... -- and most importantly **sometimes these “roads” are not biologically plausible, and yet somehow achieve great Brain-Score for better or for worse**. In our manuscript, and also in Feather et al. NeurIPS 2019, Dapello, Marques et al. NeurIPS 2020, Deza & Harrington ICLR 2022, it seems like adversarial training plays a crucial role in aligning representations of computer vision models but it may not be the only possible way.
>
> > In section 3.2: the relation with peripheral vision, the use of texform stimuli and the reasons why they are used are unclear. The first sentence of the section is misleading as your are not really interested in how well the classes are getting separated when going up in the hierarchy but whether class separation is preserved when using texform (the locally texture-matched version of the image). I think you should start this section with what is stated in the 2nd paragraph (texture-like recognition in human => mid-ventral visual cortex and texture / peripheral vision => robust perception => testing that using natural images and texform and comparing classification).
>
> **This is a great observation Reviewer 2i2E! We’ve fixed this and swapped the paragraph order!**
>
> > Font Size in figures should be close to the main text font size.
>
> Our images are saved in pdf format so you should also be able to zoom without any problems when viewed in digital format.
>
> Reviewer 2i2E, thank you for your review and suggestions. We hope to have solved your questions and motivated you to increase your score. Finally, We encourage you to read our general rebuttal regarding the novelty and significance of this work. Please let us know if you have any other questions or if something is still unclear.

---

### Official Review · Reviewer_84qC · 2022-10-24

**Confidence:** 4
**Correctness:** 3
**Technical Novelty And Significance:** 2
**Empirical Novelty And Significance:** 2
**Recommendation:** 3

**Clarity, Quality, Novelty And Reproducibility:**

The clarity can be significantly improves, especially when it comes to providing motivation for the analyses of the model representations and the significance of the findings of these analyses (see above under Weaknesses).
It is difficult for me to judge the novelty because the work needs to be better positioned in the related literature.
The quality and reproducibility of the work appear to be solid.


**Strength And Weaknesses:**

Strengths:
- investigates a timely question of why transformers are so good at predicting brain recordings
- careful evaluation of the effect of training procedure on the model representations

Weaknesses:
1. Unclear motivation for the specific investigations of the model representations, which also makes it difficult to know what to takeaway from these investigations. Why were these specific investigations chosen, and what did the authors hope to learn from each of them?
2. There is also a need to position this work more clearly with respect to the related literature. What aspects of this model are entirely new with respect to evaluating the brain score? Even if the combination of all components is novel (ViT + rotational invariance + adversarial training), were certain components previously investigated and what is the new takeaway from the current work? Also have previous works investigated the effect of these training procedures on model representations?
3. There are several points in the manuscript where the writing is difficult to follow and/or the claims are not supported by the evidence provided. A few examples:
- "interesting question that was one of the motivations of our paper: "Are Vision Transformers good models of the human ventral stream?" (how did you plan to answer this question by looking at data from non-human primates?)
- "if we find that a specific model yields high Brain-Scores, this may suggest that such a flavor of ViT-based models obey a necessary but not sufficient condition of biological plausibility" (I do not follow this logic. Perhaps you can conclude something about a sufficient condition, but I do not see anything in the experiments in this work that can suggest a necessary condition for brain alignment)
- "we observed that each step independently helps to improve the overall Brain-Score" (the observation is that each step contributes individually. The authors have not shown that the contributions are independent of each other
- Fig 3 caption: "As the average Brain-Score increases in our system, the distortions seem to fool a human as well." (is this statement based on actual human experiments or the authors' own feelings?)
- "Is this new excel in performance due to their.." (excel is a verb and not a noun)
- "their use has been carefully limited as a model of visual computation" (what does carefully limited mean?)
4. Some of the figures are illegible (Fig 1 and 2)

**Summary Of The Paper:**

This work shows that a vision transformer (ViT) model is able to achieve higher brain score (i.e. performance at predicting electrophysiology recordings of primates viewing images) than a more "biologically-plausible" alternative (a convolutional neural network + a frontal V1-like module), specifically when the ViT is trained with specific training objectives: invariance to rotation and adversarial training. The authors show that each one of these objectives individually contributes to the improvement in brain score. Additionally, this work makes some attempts to study how the representations of the well-performing ViT model change as a function of their training procedure.

**Summary Of The Review:**

While this work investigates an important and timely question of why transformer-based models are so good at predicting brain recordings, it can be significantly improved along several dimensions w.r.t. the motivation of the completed analyses and the significance of the finding, and the relation to other work in this area. In its current form, I do not believe it is ready for publication in a top-tier ML venue.

---

> ### Author Response · Authors · 2022-11-15
> **Comment response to Reviewer 84qC [1/2]**
>
> Thank you Reviewer 84qC for reviewing our paper. We will address your comments below, and encourage you to read our General Reply to all the Reviewers, which may hopefully persuade you to increase your score:
>
> > Unclear motivation for the specific investigations of the model representations, which also makes it difficult to know what to take away from these investigations. Why were these specific investigations chosen, and what did the authors hope to learn from each of them?
>
> We chose to use a dual-stream transformer that incorporates multi-scale computation and also gramian-like representations of an image. This architectural decision was justified given empirical results in visual neuroscience from Simoncelli & Freeman (Steerable Pyramids, 1995), texture-based representations of objects (Long, Yu & Konkle, 2018; Jagadeesh & Gardner, 2022; Harrington & Deza, 2022). The training optimization procedures were justified given previous research on rotation invariance (Ecker et al., 2019) and adversarial training (Santurkar, Ilyas, Tsipiras, Madry). All these decisions are properly justified in Section 2 of our initial submission. Furthermore, it is worth mentioning that **we achieved our high score in the Brain-Score competition only after our 4th entry. Several other groups (including the winner) had over 20 and in some cases 60 submissions (the winner) -- one could potentially argue that this is “overfitting wrt the competition”, while our approach was more principled.**
>
> In addition, our analysis suggests that the **optimization procedure** is really important for generating good models of the human ventral stream based on the Brain-Score Metric which quantifies the similarity of neural network responses between a computer vision model (artificial) and a set of non-human primates (biological). In that sense, we believe that understanding the importance of each step in the constrained optimization procedure of the CrossViT is of vital importance since stacking multiple data augmentations (Alexander Riedel 2022) will probably increase the “brain-likeness” but we would not understand which aspects really matters in this increase
>
> > ...What aspects of this model are entirely new with respect to evaluating the brain score? Even if the combination of all components is novel (ViT + rotational invariance + adversarial training), were certain components previously investigated and what is the new takeaway from the current work? Also have previous works investigated the effect of these training procedures on model representations?
>
> Please see our reply to your previous observation. In short: many of these phenomena have been investigated in CNN’s but not in the context of the Brain-Score competition, and also not in a principled way (with the exception of the work of Dapello, Marques et al. 2020; and recently Riedel 2022). **However no other group in the world has rigorously explored how many of these effects stack in the context of biological plausibility through Brain-Score with a Transformer-based model while at the same time breaking a record in the competition.**
>
> > "interesting question that was one of the motivations of our paper: "Are Vision Transformers good models of the human ventral stream?" (how did you plan to answer this question by looking at data from non-human primates?)
>
> As stated in our paper, “Our approach to answer this question will rely on using the Brain-Score platform (Schrimpf et al., 2020a). This platform quantifies the similarity via bounded [0,1] scores of responses between a computer model and a set of non-human primates”. Here the ground truth is collected via neurophysiological recordings and/or behavioral outputs where non-human primates are performing psychophysical tasks, and the scores are computed by some derivation of Representational Similarity Analysis (Kriegeskorte et al., 2008) when pitted against artificial neural network activations of modern computer vision models ”
>
> However, perhaps the nuance of your initial question is tied to the biological differences between non-human primates and human primates, i.e.: how will we go from having high brain-scores in non-human primates to making assertions about humans (that are also primates). First of all, the Brian-Score platform does collect the Behavioral data from humans, but for claims of correlations of V1,V2,V4, IT, **we agree that perhaps this is a stretch -- albeit a very good one**. While non-human primates and primates are different model systems, they have very similar neural representations of objects due to carry-over effects of evolution + development. In an ideal world, the Brain-Score metric would collect many of these recordings from humans (also primates) rather than non-human primates -- but this is extremely difficult (and in some cases impossible) for ethical and medical reasons. Several of these points are discussed in detail in Schrimpf’s 2018 Brain-Score paper.

---

> > ### Author Response · Authors · 2022-11-15
> > **Comment response to Reviewer 84qC [2/2]**
> >
> > > "if we find that a specific model yields high Brain-Scores, this may suggest that such a flavor of ViT-based models obey a necessary but not sufficient condition of biological plausibility" (I do not follow this logic. Perhaps you can conclude something about a sufficient condition, but I do not see anything in the experiments in this work that can suggest a necessary condition for brain alignment)
> >
> > In this case, the condition is that a model that is brain-aligned should necessarily respond very similarly to how humans or primates respond against different neurophysiological stimuli as presented in the Brain-Score platform and paper  (Schrimpf et al., 2020a). **We also believe that the brain-score metric is not a sufficient condition (as they do not tile the full image distribution in their testing dataset, and only have a small group of non-human primates from their recordings)**, this is also why we analyze other aspects as for example out-of-distribution performance, and robustness against adversarial attacks presented in our submission. We elaborate on this idea more in our Discussion & Figure 9.
> >
> > > "we observed that each step independently helps to improve the overall Brain-Score" (the observation is that each step contributes individually. The authors have not shown that the contributions are independent of each other.
> >
> > We might have missed something, but I believe we’ve shown that each step in our constrained optimization generates an increase in the Average Brain-Score compared to its pre-trained version. These effects can be appreciated in Table 2 where you can also see the impact on V1, V2, V4, IT and Behavior scores. Also, note that a value of 1 indicates that the responses from the ANN are equal to the responses of primates.
> >
> > > Fig 3 caption: "As the average Brain-Score increases in our system, the distortions seem to fool a human as well." (is this statement based on actual human experiments or the authors' own feelings?)
> >
> > You are right, these are author’s opinions, and are not from an actual psychophysical study as those done in Harrington & Deza (ICLR, 2022). However, previous works for example by Santurkar et al. (NeurIPS 2019), also suggest that these types of distortions fool humans. We agree however, that **psychophysical experiments should be done on these stimuli, and this is current follow-up work from this paper** [see for example the analysis on time-limited display of adversarial images on humans by Elsayed et al., (NeurIPS 2019)]. We have also added a footnote in the main paper regarding this question.
> >
> > > "Is this new excel in performance due to their.." (excel is a verb and not a noun)
> >
> > We changed this in our main manuscript. Thank you for your suggestion !
> >
> > > "their use has been carefully limited as a model of visual computation" (what does carefully limited mean?)
> >
> > We mean that Neuroscientists have been more careful on claiming that Transformers are “brain-inspired” as much as CNN’s (that are still loosely inspired up until V1; more than that is engineering heuristics), however, they are still used sometimes as models for research purposes in the visual neurosciences.
> >
> > > Some of the figures are illegible (Fig 1 and 2)
> >
> > Our images are saved in pdf format so you should also be able to zoom without any problems when viewed in digital format.
> >
> >
> > > The clarity can be significantly improved, especially when it comes to providing motivation for the analyses of the model representations and the significance of the findings of these analyses (see above under Weaknesses). It is difficult for me to judge the novelty because the work needs to be better positioned in the related literature. The quality and reproducibility of the work appear to be solid. --
> >
> > Thank you for this insight. I think Reviewer kkk2 (score 8 w/ high confidence) has done a great job positioning our work and its relevance in the field of ML, CV and Visual Neurosciences: https://openreview.net/forum?id=02Bt_4tx6r&noteId=ZheW01HND0b
> >
> > Furthermore please see our response to Reviewer ywsh on a very similar question regarding novelty + analysis: https://openreview.net/forum?id=02Bt_4tx6r&noteId=h-VOd9yQp57
> >
> > ----
> >
> > We overall think there is a new current of works that are at the intersection of exploring visual computation in humans and machines and both their convergences and divergences including a NeurIPS workshop called SVRHM dedicated to this theme since 2019. We think many of the questions you have raised are very important and precise as they come up in many talks and presentations. Even during this initial reviewing phase our paper has received mixed reviews (coming from reviewers with different backgrounds + angles), which in our eyes is a very good sign as we believe that our paper will foster good discussions at ICLR if it is accepted.

---

> > > ### Author Response · Authors · 2022-12-02
> > > **Rebuttal Discussion**
> > >
> > > Thank you again for your time, We have made our best to answer your questions and also follow your suggestions about clarification of supporting examples that can be seen in the last version of our submission. We would appreciate your kindly checking our response! Please, do not hesitate to contact us if there are further clarifications. Thanks!!
> > >
> > > Best wishes, Authors.

---

### Author Response · Authors · 2022-11-15
**General comments to All Reviewers**

Thank you to all reviewers for your time and effort in reading our paper. Our paper has received initial mixed reviews (both highs and lows) and we’d like to find a positive consensus (towards acceptance) below by addressing many of the main points brought up by all the reviewers.

It seems to be the case that Reviewer 84qC and Reviewer ywsh are approaching their review of our paper through a pure Machine Learning / Computer Vision lens rather than Neuro/Cog Sci which is the sub-field of expertise/area where we submitted our paper to ICLR.

For example, Reviewer 84qC has misunderstood our main claim in his/her main summary (that we will re-clarify): “This work shows that a vision transformer (ViT) model is able to achieve higher brain score (i.e. performance at predicting electrophysiology recordings of primates viewing images) than a more "biologically-plausible" alternative (a convolutional neural network + a frontal V1-like module)”. This is not our main claim or the goal of this paper. **The goal of this paper -- as our title suggests -- is that we have found a transformer-based model that when optimized a certain way can achieve a state of the art brain score in area V4 and we’d like to understand why.** It happens to be a secondary observation (as we also say in our abstract) that it achieves greater scores than the model of Dapello et al. (CNN + V1-like architecture), but this is just a happen-stance consequence that has stemmed from our main goal. We will make this clear Reviewer 84qC, thank you for your insight, and we will address many of your other points in your personalized thread below as well!

Similarly, the weakness pointed out by Reviewer ywsh (weak reject) goes in a similar direction:

> Explorations on adversarial training and robustness from Section 3.1: It would be reasonable to see the model thrive in more than the Brain-Score competition ranking when it comes to claims on better adversarial robustness.

>were there any explorations on the adversarial training pipeline (FAT, TRADES, standard AT, etc.)?

>There has been a significant line of work on exploring adversarially trained vision transformers' generalization capabilities. Can the authors compare their baseline model with some of those?

These are all great questions but are supplementary to the main goal of the paper: to dissect the contributions of the optimization procedure and the transformer-based architecture to producing a high record-breaking Brain-Score in area V4. In fact, we are exploring the answer to many of these questions pointed out by Reviewer ywsh with current experiments, but believe each one of these questions is a paper on its own if they are to be carefully examined both quantitatively and qualitatively.

Overall, we think that this work should be published in ICLR because this is where greater value for the ML, CV, and Neuroscience community will stem from concerning this type of research at the symbiotic intersection of human and machine vision with high-end ML techniques that are otherwise still lagging in the computational neuroscience & perceptual psychology community. **Furthermore, a general question that has permeated all communities mainly ML -- where neuroscience expertise is lacking -- is if Transformers in one way or another are “brain-aligned”.** We think the degree of sophistication in our computational experiments has shown “it looks like they are, even if they are not -- so let’s take a look under the hood to understand why”.

Indeed, Reviewer 2i2E (accept) positively states the paper is novel as “Vision Transformers are not yet widely tested against biological vision datasets.”, and that there is reproducibility through the Brain-Score competition. **Critically, Reviewer kkk2 who has voted 8 with a confidence of 5 signals the interdisciplinary value of our findings and how it merges the scientific axis of computer vision, ML engineering, and the underlying principles of visual neuroscience:**

> “This is an interesting paper and a valuable contribution in that it should spark an important discussion on how we should go about assessing brain-likeness of an ANN architecture. To me the most important message from this paper is, as I've long suspected, that if you fiddle with any architecture long enough then you can make it score high on Brain-Score. We, as a community, should be much more careful when claiming brain-likeness based on Brain-Score, and this paper is an important voice in this discussion.

> The paper is well-written, the experiments support the claims and those claims are important and/or novel.

> My recommendation is to accept this work as I would definitely like to see it presented at the conference.”

We hope the AC, and reviewers take into account these thoughts and are inclined to have this paper accepted at ICLR which will encourage further discussions on this topic and will be of value to the CV, ML & Neurosciences communities.

---

### Decision · Program_Chairs · 2023-01-20

**Decision:**

Reject

**Justification For Why Not Higher Score:**

The significance of the results is limited to a small community of computational neuroscientists interested in maxing out the performance of models on brainscore.

**Justification For Why Not Lower Score:**

NA

**Metareview: Summary, Strengths And Weaknesses:**

This work shows that a vision transformer model can achieve a high brain score (brain score is a benchmark used to evaluate the accuracy of SOTA models to predict monkey electrophysiology data) after it is trained with data augmentation to yield invariance to rotation and adversarial training. There is general consensus that the work is well done but there was some initial disagreement regarding the value of the work for either ML or neuroscience.

**Summary Of Ac-Reviewer Meeting:**

 We met with the reviewers and after discussion there was broad agreement that the work appeared of limited significance to the neuroscience community because it does not propose any specific mechanism to be tested in the brain (data augmentation or adversarial training are not things that can be tested experimentally). In fact, at the meeting it became clear that one of the 2 positive reviewers somehwat misunderstood the message of the paper. The reviewer thought that the paper was about cautioning researchers about the fact that a kitchen sink ML approach can yield SOTA results on brainscore underscoring the limitations of brainscore (!) when in fact the message of the paper is that transformers are good models of primate vision. Overall there was general agreement that the contributions of the paper were quite limited.